# An Unsupervised Error Detection Methodology for Detecting Mislabels in Healthcare Analytics

**DOI:** 10.3390/bioengineering11080770

**Published:** 2024-07-31

**Authors:** Pei-Yuan Zhou, Faith Lum, Tony Jiecao Wang, Anubhav Bhatti, Surajsinh Parmar, Chen Dan, Andrew K. C. Wong

**Affiliations:** 1Department of Systems Design Engineering, University of Waterloo, Waterloo, ON N2L 3G1, Canada; faith.lum@uwaterloo.ca (F.L.); jiecao.wang@uwaterloo.ca (T.J.W.); akcwong@uwaterloo.ca (A.K.C.W.); 2AI Engineering Team, SpassMed Inc., Toronto, ON M5H 2S6, Canada; anubhav.bhatti@spassmed.ca (A.B.); suraj.parmar@spassmed.ca (S.P.); chen.dan@spassmed.ca (C.D.)

**Keywords:** unsupervised learning, error detection, pattern discovery and disentanglement, healthcare data analysis

## Abstract

Medical datasets may be imbalanced and contain errors due to subjective test results and clinical variability. The poor quality of original data affects classification accuracy and reliability. Hence, detecting abnormal samples in the dataset can help clinicians make better decisions. In this study, we propose an unsupervised error detection method using patterns discovered by the Pattern Discovery and Disentanglement (PDD) model, developed in our earlier work. Applied to the large data, the eICU Collaborative Research Database for sepsis risk assessment, the proposed algorithm can effectively discover statistically significant association patterns, generate an interpretable knowledge base for interpretability, cluster samples in an unsupervised learning manner, and detect abnormal samples from the dataset. As shown in the experimental result, our method outperformed K-Means by 38% on the full dataset and 47% on the reduced dataset for unsupervised clustering. Multiple supervised classifiers improve accuracy by an average of 4% after removing abnormal samples by the proposed error detection approach. Therefore, the proposed algorithm provides a robust and practical solution for unsupervised clustering and error detection in healthcare data.

## 1. Introduction

Machine Learning (ML) models are used to process big healthcare datasets, and various ML models have proven their effectiveness for predictive analytics and diagnostic applications [1]. For instance, Alvin et al. [2] demonstrate how deep learning models improve healthcare quality when applied to electronic health records (EHR) data. Usharani et al. [3] introduced a specialized Fuzzy C-mean segmentation model to improve brain tumor identification. Similarly, another deep learning model is used for rectal cancer detection [4]. In [5], classical and deep learning models were used to estimate the compensatory index for hypovolemia, enhancing emergency medicine assessment. In [6], an LSTM was developed for early diabetes detection. However, there are several challenges that machine learning may face in healthcare data analysis [7].

First, machine learning models always require accurate and complete datasets to ensure effective functionality. Thus, data quality significantly influences reliability and can impact clinical decision-making and patient care [8]. Specifically, the reliability of labels in healthcare data is crucial for the effectiveness of predictive models. When class labels are inaccurate or unreliable, even models with high predictive accuracy may not be reliable due to their dependence on incorrectly labelled data. Moreover, it is difficult for medical experts to manually detect these elusive errors due to lack of contextual information, limiting data privacy regulations, and the sheer scale of data to be reviewed [9].

Next, healthcare providers often need clear explanations for AI-driven decisions, which is challenging with complex models like deep neural networks. Many of these models are black box models, such as genetic algorithms solely focusing on performance [10], lacking transparency and accountability, which can lead to severe consequences [11]. For high-stake applications such as healthcare, the judicial system, and human resource recruitment, machine learning models must be transparent to enable the interpretation of their predictions and explanation of decision boundaries [11,12,13]. As suggested by Rudin [11], creating an interpretable model is a better solution than trying to explain black box models. This allows for synergistic improvement as the interpretable model both integrates and enhances expert knowledge.

Finally, many medical conditions are rare, resulting in imbalanced datasets where the number of cases for one outcome is much smaller than those for others. This class imbalance problem [14] has posed a challenge for years, with machine learning models still struggling to overcome it.

To address these challenges, we have developed a model based on statistical theory, named Pattern Discovery and Disentanglement (PDD) [15,16], to efficiently discover compact sets of significant association patterns linked to their primary sources. It generates a knowledge base interlinking patterns, entities, and sources for visualization, interpretation, and further exploration. In this paper, we propose an unsupervised error detection method using patterns discovered by the PDD to improve the quality of the data.

Some existing algorithms also focus on interpretable patterns. For example, patterns from four gene expressions were explored to classify patients effectively [17], but the number of explored patterns is relatively small, considering the initial input containing 23 features. However PDD’s explainability is more succinct, direct, and extensive, which can use the full dimensions of its input. In addition, the local rule-based explainability method described in Metta et al. [18] uses a genetic algorithm to create a synthetic neighborhood upon which an interpretable model such as a decision tree is applied. This method is limited by the scope of rules that can be extracted from models like decision trees, which are relatively simple and outperformed by PDD.

To identify errors, existing error correction strategies often rely on prediction results from a trained model. These strategies include those implemented through Cleanlab, isolation forest, and AdaBoost. Cleanlab’s system [19] can provide model-specific insights, which are highly dependent on the model’s quality and may inherit biases. Isolation Forest can be used for anomaly detection by determining which points require fewer partitions to separate from other data points [20]. AdaBoost calculates the weighted error of the classifier for each iteration, giving higher weights to misclassified samples to emphasize their importance in the next iteration [21]. Unlike the above strategies, our proposed error detection approach does not depend on a specific model and avoids the issue of model bias. Moreover, it provides an interpretable explanation of dataset and error properties, ensuring a higher-quality dataset before any model training occurs.

To demonstrate the effectiveness of the proposed algorithm, we applied it to the eICU Collaborative Research Database (eICU-CRD) [22] for assessing sepsis risk based on vital signs. Sepsis is a condition that requires prompt diagnosis [23]. Our analysis confirms that the algorithm successfully addresses imbalances in healthcare data, identifies errors and mislabels, and enhances sepsis detection.

Therefore, the contributions of this paper are significant both algorithmically and practically. Algorithmically, (1) we introduce an unsupervised algorithm that effectively discovers patterns or groups of patterns associated with distinct classes, which are functional and statistically independent from the patterns of other classes, as demonstrated by our experimental results; and (2) we propose an error detection method to detect mislabels in the dataset, the effectiveness of which is validated through comparing classification results before and after error correction. Practically, (1) the application of our algorithm to the disease sepsis reveals insightful characteristics of patients and patient groups, enhancing our understanding of the disease’s impact; and (2) the error detection process improves the quality of the dataset, helping to identify outliers or mislabelled patient records.

## 2. Materials and Methods

### 2.1. Dataset and Preprocessing

The eICU-CRD [22] includes data from critical care unit patients across the United States for 2014 and 2015. We used the same dataset as the study mentioned in [23], which was collected from key tables such as patient demographics, diagnoses, nursing assessments, and vital periodic records. This reference dataset consisted of 138 features related to patient information, baseline measurements, labels, and temporal vital signs. There are features related to patient data, baseline physical measurements, diagnosis, label, and temporal vital signs. Patient data includes parameters like gender, age, ethnicity, weight and height. Baseline physical measurements include Glasgow coma scale (GCS), systolic blood pressure, diastolic blood pressure, mean blood pressure, pulse pressure, heart rate, respiration, and SpO2 oxygen saturation. Diagnosis includes diagnosisoffset and diagnosispriority. Label includes ICD9 codes and categories. Temporal vital sign data includes feature-lagged physical measurements.

To clean up the dataset, we removed 12 features with excessive missing values, reducing the feature count from 138 to 126. Further preparing the dataset for analysis and preventing data leakage, we excluded an additional six label-related features (e.g., ICD9, categories 1–4, and binary label) and six features used solely for preprocessing (e.g., patientunitstayid, observationoffset, activeupondischarge, diagnosisoffset, diagnosispriority, hospitaladmitoffset). Labels were generated from clinician-evaluated ICD9 codes corresponding to specific diagnoses, and the records were categorized into two groups: *sepsis* and *other*. It was notably imbalanced, with 24.4% of the samples being sepsis-positive [23]. The final dataset used to compare PDD’s error detection comprised of 10,743 records, 113 features, and one label feature. Notably, while the full 114-feature dataset was compared before and after error detection using supervised learning models in Section 3.4, only a 28-feature subset was used by PDD to interpret the dataset and compare unsupervised learning methods in Section 3.1 and Section 3.2. Detailed processing steps are included in the Appendix A with relevant features defined in Table A1.

### 2.2. Methodology

In this section, we presented the proposed algorithm applied to the dataset descriptive in Section 2.1. The algorithm completes three main tasks: (1) Pattern discovery via PDD for interpretability, as discussed in the [16], (2) Unsupervised learning applied to the dataset; (3) Detecting errors or abnormal samples (i.e., mislabelled records, outliers, and undecided records).

#### 2.2.1. Interpreting Dataset

Our early work with PDD focused on interpretable pattern discovery. In this section, we presented how PDD works for interpretability. As detailed in Appendix A, the processed dataset of 114 features will be used to compare the supervised learning models. This dataset includes features related to patient information, physical assessments, and temporal features capturing two observations (i.e., time points 2 and 14) of six physical measurements. Statistics such as mean, standard deviation, minimum, maximum, kurtosis, and skewness are calculated for each observation. In the pattern discovery process, we aim to discover associations between feature values. Given that many of the statistical features convey the same information, to simplify the analysis and enhance interpretability, we only retained the standard deviation for each observation, as it best describes the variability of data. Therefore, to avoid the complexity introduced by an overwhelming number of features (attributes), we finally retained a subset of 27 features describing patient demographics and their physical assessments. This includes 15 features related to physical measurements and patient information (i.e., GCS, systolic blood pressure, diastolic blood pressure, mean blood pressure, pulse pressure, heart rate, respiration rate, SpO2, age, gender, ethnicity, discharge status, admission weight, discharge weight, and height); another two observation features for each of the six physical measurements, totaling 12 features; and one label feature. Consequently, the dimension of the dataset is 10,743 by 28.

Due to the unlimited degrees of freedom inherent in numerical features, correlating these features with the target variable and interpreting the associations present significant challenges. So, the first step of the PDD process is discretizing the numerical features into event-based or discrete categories according to clinical standards as Table 1 shows. Other numerical features without clear clinical significance such as age and admission weight, were discretized into intervals using the Equal Frequency [24]. This method ensures an equal distribution of data points (i.e., records) within each interval. It is achieved by first sorting the numerical data in ascending order, and then dividing the data into three intervals (bins) each containing an equal number of data points. In information theory, this method indirectly maximizes the data’s informational content. It also handles skewed distributions and simplifies analysis with interpretable categories that are more understandable than precise numerical values. Hence, by applying this discretization approach, all features are converted to categorical ones.

Then, on the discrete dataset, using PDD [16], we construct a statistical residual matrix (SR-Matrix) to account for the statistical strength of the associations among feature values. In pattern discovery, the term “attribute” is used instead of “feature”, so “attribute value” (AV) will be used subsequently. Since the meaning of the attribute values (AVs) and their class labels are implicit, the discovery of a statistically significant association of the AVs is unaffected by prior knowledge or confounding factors. To evaluate the association between each pair of AVs in the SR-Matrix, we calculated the statistical measure of adjusted standardized residual to represent the statistical weights of the association between distinct AV pairs. For instance, if $AV_{1}$ represents the value of attribute systolic as *H*, and $AV_{2}$ represents the value of attribute diastolic as *L*. Then the adjusted standard residual of the association between $AV_{1}$ and $AV_{2}$ is calculated in Equation (Equation 1), which is denoted as $SR(AV_{1},AV_{2})$.
(1)$$SR\left(AV_{1},AV_{2}\right)=\frac{Occ\left(AV_{1},AV_{2}\right)-Exp\left(AV_{1},AV_{2}\right)}{\sqrt{Exp\left(AV_{1},AV_{2}\right)\cdot \left(1-\frac{Occ(AV_{1})}{N}\right)\cdot \left(1-\frac{Occ(AV_{2})}{N}\right)}}$$
where $Occ(AV_{1})$ and $Occ(AV_{2})$ represent the number of occurrences of each attribute value; $Occ(AV_{1},AV_{2})$ is the total number of co-occurrences of the two attribute values; and $Exp(AV_{1},AV_{2})$ refers to the expected frequency of co-occurrences of the two attribute values; *N* is the total number of entities. Further details on the calculation are provided in Appendix B.

The key idea of the pattern discovery and disentanglement are briefly described below. We assume that certain attribute values (AVs) or associations co-occur on samples due to certain primary causes (referred to as primary sources). Other significant or insignificant AV associations may also occur in the samples, entangled with those originating from the primary sources. Hence, the goal of PDD is to discover statistically significant association patterns from AV groups on a disentangled principal component (PC), then cluster each AV group into subgroups, denoting them as Disentangled Statistical Units (DSUs). These DSUs are functionally independent and statistically connected within the same group but not associated with other groups.

To achieve such a goal, PDD first applies a linear transformation, Principal Component Analysis (PCA), to decompose the SR-Matrix into Principal Components (PCs). Each PC is functionally independent, capturing associations uncorrelated to those obtained in other PCs. To look for AVs with statistically significant associations with other AVs on the same PC, PDD reprojects each PC back onto an SR-Matrix to generate a Reconstructed SR-Matrix (RSR-Matrix) for each distinct PC. If the maximum residual between a pair of attribute values (AVs) within an RSR-Matrix exceeds a statistical threshold, such as 1.96 for a 95% confidence interval, the association is considered statistically significant. Notably, the associations identified within each RSR-Matrix (or PCs) remain functionally independent from those in other RSR-Matrices (or PCs). Then, on the PCs with statistically significant associations, we obtain AV groups where the AVs within are statistically connected. From entities containing these AVs, we use a hierarchical clustering algorithm to obtain entity clusters containing AV subgroups with statistically connected AVs within, but not outside of that subgroup. They are the DSUs originate from distinct primary sources.

Figure 1 illustrates the AV association disengagement concept. After Principal Component Decomposition of the SR-Matrix, PCs and their corresponding RSR-Matrices are obtained. Only RSR-Matrices containing SR values exceed the statistical significance threshold (i.e., 1.96), and their corresponding PCs are retained. For each of those retaining PCs, three groups of projected AVs can be identified along its axis, each showing different degrees of statistical associations.

Those close to the origin are not statistically significant and thus do not associate with distinct groups/classes (marked as (a) in Figure 1), They are close to the origin because all their coordinates (SRs) are low and insignificant.Those at one extremal end of the projections; and those at the opposite extremal end (both marked as (b)).The AV groups or subgroups in (b), if their AVs within are statistically connected but disconnected from other groups, may associate with distinct sources (i.e., classes) (marked as (c)).

As a result, two AV groups at opposite extremes were discovered. Each AV within such a group is statistically linked to at least one other AV within, and none of them is statistically connected to AVs in other groups. Furthermore, it is possible that among the AVs in the AV groups, some of the AV Subgroups may only occur on subsets of the entities uncorrelated to other subgroups. Hence, to achieve a more detailed separation of groups, several subgroups are separated in each AV group based on their appearance in entity groups. This is done using a similarity measure defined by the overlapping of entities each AV can cover. Then AV subgroups found in different entity groups can be identified. We denote such an AV subgroup by a three-digit code [*#PC, #Group, #SubGroup*] and refer to it as a Disentangled Space Unit (DSU). We hypothesize that these DSUs originate from distinct functional sources.

Therefore, in each subgroup denoted by DSU, a set of AVs is included, which are referred to as pattern candidates. We then developed a pattern discovery algorithm to grow high-order patterns, called comprehensive patterns, from the pattern candidates. In the end, a set of high-order comprehensive patterns is generated within each DSU, and they are all associated with the same distinct source.

The interpretable output of the PDD is organized in a PDD Knowledge Base. This framework is divided into three parts: the Knowledge Space, the Pattern Space, and the Data Space. Firstly, the Knowledge Space lists the disentangled AV subgroups referred to as a Disentangled Space Unit (DSU) (denoted by a three-digit code, [*#PC, #Group, #SubGroup*] shown in the three columns of the knowledge space to indicate different levels of grouping) linking to the patterns discovered by PDD on the records. Secondly, the Pattern Space displays the discovered patterns, detailing their associations and their targets (the specified class or groups). Thirdly, Data Space shows the record IDs of each patient, linking to the knowledge source (DSU) and the associated patterns. Thus, this Knowledge Base effectively links knowledge, patterns, and data together. If an entity (i.e., a record) is labelled as a class, we can trace the “what” (i.e., the patterns it possesses), the “why” (the specific functional group it belongs to), and the “how” (by linking the patterns to the entity clusters containing the pattern(s)).

The novelty and uniqueness of PDD lie in its ability to discover the most fundamental, explainable, and displayable associations at the AV level from entities (i.e., records) associated with presumed distinct primary sources. This is based on robust statistics, unbiased by class labels, confounding factors, and imbalanced group sizes, yet its results are trackable and verifiable by other scientific methods.

#### 2.2.2. Clustering Patient Records

Without specifying the number of clusters to direct the unsupervised process, PDD can cluster records based on the disentangled pattern groups and subgroups.

As described in Section 2.2.1, the output of PDD is organized into a Knowledge Base, where each pattern subgroup is represented by a DSU[#PC,#Group,#SubGroup]. As defined in Section 2.2.1, the set of AVs displayed in each DSU is a summarized pattern, representing the union of all the comprehensive patterns on entities discovered from that subgroup. We denote the number of comprehensive patterns discovered from the summarized pattern in DSU as #CP−DSU. For example, in DSU[1,1,1], if 10 comprehensive patterns are found, then #CP[1,1,1]=10. Each record may possess none, one, or multiple comprehensive patterns for each DSU. We denote the number of comprehensive patterns possessed by a record in a specific DSU as #ID−DSU. For example, #1[1,1,1]=5 and #1[2,1,1]=6 represent the record with ID=1 possess 5 comprehensive patterns in DSU[1,1,1] and 6 comprehensive patterns in DSU[2,1,1].

Each DSU can represent a specific function or characteristic in the data, potentially associated with a particular class. For example, in this study, DSU[1,1,1] is associated with sepsis, while DSU[1,2,1] is associated with other. The fact that DSU[1,1,1] and DSU[1,2,1] appear as two opposite groups in PC1 (Figure 1) indicates that their AV associations have significant differences as captured by PC1. Some DSUs might reveal rare patterns not associated with any class while the class label is not in the association.

Based on the definitions described above, we cluster the records by assigning each record to the class that matches the most comprehensive patterns compared to any other class. To provide a more detailed explanation of the clustering process, consider the following example. The DSU outputted by PDD are DSU[1,1,1], DSU[1,2,1], DSU[2,1,1], and DSU[2,2,1], which are associated with sepsis, other, sepsis, and other, respectively. The total number of comprehensive patterns in these DSUs are #CP[1,1,1]=100, #CP[1,2,1]=100, #CP[2,1,1]=200, and #CP[2,2,1]=200 respectively. Consider a record (ID=n) with comprehensive patterns possessed by this record are #n[1,1,1]=50, #n[1,2,1]=60, #n[2,1,1]=0, and #n[2,2,1]=150.

Due to the variation in the number of comprehensive patterns across DSUs, we use a percentage rather than an absolute value to measure the association of the record with pattern groups. Hence, to determine how the record (ID=n) is associated with the sepsis patterns, we calculate the average percentage of the number of comprehensive patterns associated with a specific class possessed by the record, denoted as #ID−Class. Due to #n[2,1,1]=0 indicates that the record is not covered by the DSU[2,1,1], it is excluded from the calculation to avoid the significant impact of a zero value on the final percentage. Hence, the association of the record (ID=n) with the sepsis patterns is calculated as #n−sepsis=50/100=0.5. Similarly, #n−others=mean(60/100,150/200)=0.675. Since #n−others is greater than #n−sepsis, the record is assigned as other. To evaluate the accuracy of this assignment for all records, we compare the assigned class label with the original implicit class label.

#### 2.2.3. Detecting Abnormal Records

The evaluation of the classification or prediction involves comparing the predicted labels with the original class labels. However, this comparison is unreliable if mislabels exist in the original data. To address this issue, we proposed an error detection method to identify abnormal records using the patterns discovered by PDD. In our early work on PDD, we integrated both the supervised and unsupervised methods for error detection and class association. In this paper, we simplify the process by using only a novel unsupervised method on a dataset with implicit class labels as the ground truth, making the error detection process more succinct.

To determine whether a record is abnormal, the proposed algorithm compares the class assigned by PDD with its original labels, evaluating the consistency of discovered patterns with their respective explicit or implicit class labels. We define three statuses to an abnormal record: **Mislabelled**, **Outlier**, and **Undecided**, which are detailed below.

**Mislabelled**: If a record is categorized into one class but matches more patterns from a different class according to the PDD output, it suggests the record may be **mislabelled**. For example, consider the same record with ID=n described in Section 2.2.2 with the same setting of the pattern groups where #n−sepsis=0.5 and #n−others=0.675. If the record is originally labelled as sepsis in the dataset, but the relative difference (|#n−others−#n−sepsis||#n−others+#n−sepsis|) is greater than 0.1, this suggests that the record (ID=n) is more associated with other than with sepsis. The relative difference is used instead of absolute difference because it provides a scale-independent comparison of the number of patterns associated with one class to another. A value greater than 0.1 indicates that the number of patterns associated with one class is statistically significantly greater than the number associated with another class. Hence, the record (ID=n) may be **mislabelled**.**Outlier**: If a record possesses no patterns or very few patterns, it may indicate the record is an **outlier**. For example, a record with ID=m uses the previously described pattern group settings. The comprehensive patterns possessed by this record are: #m[1,1,1]=1, #m[1,2,1]=0, #m[2,1,1]=0, and #m[2,2,1]=1. Calculating the percentages, #m−sepsis is 1/100=0.01 and #m−others is 1/200=0.005. Both percentages are less than or equal to 0.01, suggesting that record *m* possesses fewer than 1% of the patterns associated with either class, which may indicate it is an **outlier**.**Undecided**: If the number of possessed patterns for a record is similar across different classes, the record should be classified as **undecided**. For example, a record with ID=k uses the previously described pattern group settings. The comprehensive patterns possessed by this record are: #k[1,1,1]=50, #k[1,2,1]=60, #k[2,1,1]=110, and #k[2,2,1]=100. Calculating the percentages, #k−sepsis is the mean of 50/100 and 110/200, which is 0.55; and #k−others is the mean of 60/100 and 100/200, which is 0.55. Since the difference between the two percentages is zero or less than 0.001, record *k* may be associated with both classes, suggesting it is **undecided**

To avoid adding new incorrect information, mislabelled, undecided, and outliers are removed from the dataset. Hence, to validate the effectiveness of abnormal records detected by PDD, we compared classification results from the original dataset to those from a dataset without abnormal records when various classifiers were applied.

## 3. Results

The experimental results are presented in this section. First, Section 3.1 demonstrates the interpretable output derived from PDD. Next, Section 3.2 compares PDD’s clustering accuracy with the K-means algorithm which serves as our baseline. Then, Section 3.3 presents our error detection results, showing interpretable outcomes with discovered patterns after mislabeling detection. Finally, Section 3.4 demonstrates the efficacy of error detection by comparing the performance of supervised learning algorithms both before and after the application of error detection.

### 3.1. Interpretable Results

In this section we present interpretable results, via the PDD knowledge base, containing all summarized patterns from different DSUs in Figure 2. The knowledge base contains three parts: the knowledge space, the pattern space, and the data space. First, the Knowledge Space uses the triple code of the DSU to indicate the primary source that gives rise to the disentangled patterns from the AV Subgroup. In this specific study, as shown in Figure 2, the first row displays the group discovered as the first disentangled space denoted by the code [1,1,1], indicating in reverse order the first Sub-Pattern group (SubGroup) of the first Pattern Group (Group) discovered in the first Principal Component (PC). Here, we identify two groups corresponding to two types of diseases: sepsis and other. The results show all DSUs discovered by PDD representing all the disentangled spaces with statistically significant patterns obtained from the first three PCs.

Second, the Pattern Space reveals the specific AV association patterns and their targets (the specified class or groups, if given) obtained from the DSU. We find that the disease *sepsis* exhibits different patterns than *other*. For example, the *sepsis* class was associated with lower blood pressure, higher heart rate, higher respiration, and moderate or severe GCS. Inversely, the *other* class was associated with higher blood pressure, normal heart rate, lower respiration, and mild GCS. Since the patterns outputted by PDD are disentangled, they clearly distinguish between the different classes of *sepsis* and *other*.

The patterns related to sepsis demonstrate clinical relevance as hypotension, tachycardia, elevated respiratory rate, and altered mental status are all considered key criteria defined in the Third International Consensus for Sepsis and Septic Shock (Sepsis-3) [25]. Singer et al. [25] discussed how septic shock is associated with underlying circulatory, cellular, and metabolic abnormalities that substantially increase mortality. These characteristic abnormalities helped to improve understanding of sepsis pathobiology and informed treatment such as a vasopressor requirement to constrict blood vessels and maintain a minimum mean arterial pressure in the absence of hypovolemia or adequate fluid resuscitation. This shows evidence of a positive correlation between low blood pressure and the severity of sepsis among patients with septic shock, which is consistent with patterns discovered by PDD. In addition to this, sepsis is impacted by various biomarkers and physiological parameters, such as C-reactive protein, white blood cell count, and central venous pressure [26,27]. Thus, with a more detailed dataset, PDD can further examine associations between sepsis and these factors.

Thirdly, the Data Space displays the records with their IDs from the original dataset and the discovered patterns each record possesses. As shown in Figure 2, the number in the column of each ID on the row linking to the pattern(s) and the DSU(s) represents the number of the comprehensive patterns in a DSU that the record possesses. For example, 180 indicates there are 180 comprehensive patterns in the DSU[1,1,1] covered by record 1. Due to space limitations, only the first nine records (ID=1to9) are displayed.

In the original data, records labelled sepsis are listed before those labelled other, so the first nine records belong to the sepsis class. It is also evident in the Data Space that the records possess more patterns associated with sepsis than with other.

### 3.2. Comparison of Unsupervised Learning

In addition, we compared the PDD’s clustering results with the classical clustering algorithm K-mean as the baseline. For a clear comparison, both the same subset (10,743×28) and the full dataset (10,743×114) were used to evaluate K-means performance. To run K-means, we used the sklearn.clusters package in Python 3.0 [28] with all default parameter settings and assigned the number of clusters as two, as the original data only contains the two classes of sepsis and other.

To evaluate an imbalanced dataset, balanced accuracy is more reliable than regular accuracy. Using regular accuracy, a model might frequently predict the majority class and still achieve a high accuracy rate, despite failing to accurately predict the minority class [29]. Balanced accuracy, however, considers the true positive rate of each class, offering a more honest evaluation of model performance, which is essential when one class significantly outnumbers another. Therefore, we utilize both balanced accuracy and accuracy for evaluating classification or clustering performance, in addition to precision, recall, and a weighted F1-score for comparison.

Based on the unsupervised clustering results in Table 2, PDD significantly outperformed K-Means on both the 28-feature subset and 114-feature dataset. On the 28-feature subset, the balanced accuracy of 0.89 achieved with PDD was significantly higher than the 0.42 achieved with K-means. On the 114-feature dataset, K-means exhibited improved performance, demonstrating the benefit of including more features. However, even when comparing PDD’s performance on the reduced feature subset to K-means performance on the full feature dataset, PDD still outperformed K-means by 38%.

The two clustering methods were also compared in terms of their effectiveness in handling dataset imbalance, given that only 24.4% of samples were sepsis-positive. As shown in Table 2, PDD’s balanced accuracy on the 28-feature subset exceeds regular accuracy indicating a more effective prediction of the minority class sepsis. In contrast, K-means’ regular accuracy exceeds balanced accuracy on both the 28 and 114-feature datasets indicating a more effective prediction of the majority class other. Moroever, the difference between the accuracy and balanced accuracy of PDD results is a minimal 2% in comparison to K-means’ results with differences of 10% and 24%. This indicates that PDD is better suited for handling imbalanced data and is a valuable tool for datasets where accurate prediction of the minority class is crucial. Also, that PDD has learned meaningful patterns in the minority class despite its scarcity in the dataset while K-means struggles to accurately capture these characteristics and is therefore challenged by the imbalanced distribution. In this case, PDD not only predicts more accurate clustering results for the minority class but also provides patterns that reveal the characteristics of the cluster. PDD can display all the statistically significant patterns with their source (DSU) possessed by each cluster. It also corrects mislabels based on its error detection capability and the records’ implicit class labels.

### 3.3. Error Detection Results

This section presents the error detection results using the PDD knowledge base. As mentioned in Section 2.2.3 and Section 3.1, we used the comprehensive pattern count from the PDD data space to detect abnormal records in the original dataset. For example, ID=5510 has #5510[1,1,1]=30, #5510[2,1,1]=13, #5510[3,1,1]=2, #5510[5,1,1]=6, and all DSU is associated with sepsis. Thus, #5510−sepsis=mean(30/1357,13/2617,2/453,6/940)≈0.0095. Similarly, #5510[1,2,1]=2, #5510[2,2,1]=51, #5510[3,2,1]=3, #5510[5,2,1]=8, thus #5510−other=mean(2/696,51/2514,3/561,8/1712)≈0.0083. Both #5510−sepsis and #5510−other are less than the threshold 0.01 specified in Section 2.2.3, ID=5510 is categorized as an outlier.

In total, PDD identified 996 mislabelled records, 2 outliers, and 159 undecided records. To avoid adding new incorrect information, we finally removed the 1157 records, and 9586 records were retained. For simplicity, only a partial result is shown in Figure 3.

For better visualization, in Figure 3a, all the DSUs with label=sepsis are highlighted in red, and all the DSUs with label=other are highlighted in green. An important observation after such highlighting is that attributes are consistent within its class. For example, for all DSUs with label=sepsis, their blood pressure features (i.e., systolicbp, diastolicbp, meanbp) are mostly low. And for all DSUs with label=other, their blood pressure attributes are mostly high. Due to this consistency, we assigned the same highlighting scheme to specific AV in Figure 3b. For example, when systolicbp=low, we highlighted the AV in red, implying its positive correlation with sepsis through PDD’s Pattern Space. With such a highlighting scheme, one can easily observe the pattern entanglement among the original dataset and how well PDD can disentangle the source attributes with a statistically supported, interpretable knowledge base.

### 3.4. Error Detection Applied to Supervised Learning Models

To demonstrate the effectiveness of the error detection methods, we compared the classification results of various classifiers on the dataset before and after removing errors. The classifiers tested include Random Forest, SVM, Neural Networks, Logistic Regression, LightGBM, and XGBoost. 80% of the data are used for training and 20% for testing. To evaluate the results, the same metrics (i.e., Precision, Recall, Accuracy, Balanced Accuracy, and Weighted F1-Score) are used for comparison.

Table 3 shows the comparison results of existing ML models before and after error detection. This error detection process by PDD resulted in an accuracy improvement of approximately 1–9% and 4% on average. The error detection results show how PDD can remove those abnormal records (outliers/undecided/mislabelled) to improve the accuracy of class association for building a classifier based on a cleaner training set since the balanced accuracy is increased for all classifiers.

In Table 3, it is also interesting to note how different classifiers handle overfitting and bias in imbalanced data with varying capabilities. First, for all classifiers, the regular accuracies are greater than the balanced accuracies, indicating that the predictions are more effective for the majority class, particularly for the first four traditional classifiers (RF, SVM, NN, and LR). However, for models with class balancing capability (LightGBM and XGBoost), the differences between regular accuracy and balanced accuracy are small. Second, the higher accuracy before error detection indicates that models are performing better on the majority class, which can be a sign of overfitting to the majority class. So the accuracy slightly decreases after removing abnormal cases due to reducing the overfitting for the majority class. Lastly, after error detection, the balanced accuracies increase. This indicates that outliers and undecided cases were biasedly absorbed into the majority classes before error detection, but after error detection, the classifiers show better prediction results for the minority class, leading to better results for the balanced prediction.

Additionally, to effectively highlight comparison results, Figure 4 displays radar charts that visualize each classifier’s overall performance profile. These charts distinctly show that the blue areas, representing improved metrics, are larger than the red areas for all models, with particularly obvious differences in LightGBM and XGBoost.

## 4. Discussion

In this section, we discuss the results of interpretability and error detection using PDD from the perspective of previous studies.

### 4.1. Interpretability

Decision Trees are widely regarded as interpretable models, and similarly, Random Forests (RF) can also provide interpretable insights despite being less straightforward due to their ensemble nature. Various methods have been developed to extract feature importance or feature contributions to enhance its interpretability [30]. In this section, we compare the interpretability between RF and PDD.

First, according to the RF model illustrated in Figure 5, RF assesses the importance of features associated with class labels. Similarly, as shown in Figure 2, PDD discovers patterns that reveal the associations between feature values rather than just feature associations.

Second, the importance analysis provided by RF shows the association between one feature and the target, while PDD goes further by revealing the significance of high-order comprehensive patterns linked to the target, which provides a more direct, succinct, precise, and interpretable understanding of the problem. This capability allows PDD to provide a more in-depth and precise interpretation, which is crucial for high-stakes applications requiring transparency and explainability.

Lastly, RF can also output prediction rules similar to the high-order patterns in PDD. However, according to the supervised learning comparison results provided in Section 3.4, before error detection, the accuracy of Random Forest is 86%, while the balanced accuracy is only around 73%. This indicates that, for imbalanced classes, the rules outputted by RF are easily biased towards the majority class. In contrast, as demonstrated in the study [16], PDD can discover disentangled patterns even for imbalanced data. Furthermore, PDD uses hierarchical clustering to highlight the independent functions of each pattern group, enhancing interpretability. It generates a knowledge base linking patterns, entities, and classes, which can trace the origins of the discovered patterns. This capability ensures the accuracy of error detection. This level of detail and traceability is a limitation of the RF model.

### 4.2. Error Detection

Real-world data may contain inconsistencies due to incorrect data collection, human errors, equipment inaccuracies, and missing values [31]. So removing errors and cleaning data is essential in developing a reliable ground truth for training and testing. In this study, the conditions of patients with sepsis may be various, leading to fluctuations that complicate establishing baseline values. The eICU-CRD dataset, collected in 2014 and 2015, predates the sepsis definition updated in 2016 [25], with the initial definition in 2001. Thus, the labels in the dataset may not reflect the latest medical insights. Furthermore, integrating data from various hospitals with different environments, procedures, equipment, and staff may introduce non-standardized variables.

PDD’s error detection results in Section 3.4 demonstrate removing abnormal data and creating a more reliable ground truth can improve model performance. The interpretability of PDD enables it to discover and disentangle patterns, which can be used to assign status—mislabelled, outlier, and undecided—to records. The traceability of PDD allows humans to trace patterns back to the data and clarify why a particular sample should be corrected. Furthermore, unlike other existing error detection methods that rely on model predictions, PDD is model-independent.

To better understand PDD’s error detection, sample scatter pair plots between the features of the eICU-CRD dataset are shown in Figure 6. The outlier, undecided, and mislabelled records are the 10 abnormal records presented in Section 3.3. In Figure 6a,b, regardless of whether the x-axis values are continuous or discrete, the decision boundaries for classification are not linear. Two classes are mixed, making classification and error detection based on the original features challenging due to entanglement.

However, the two classes can be visually disentangled in PDD’s Data Space as shown in Figure 7 and Figure 8 by treating each DSU as a distinct feature and the comprehensive pattern count as the feature value. The only difference between Figure 7 and Figure 8 is the proportion of the abnormal records that are highlighted. Figure 7 highlights the 10 abnormal records from Section 3.3. Figure 8 highlights all the 1157 abnormal records, which comprise 996 mislabelled, 2 outliers, and 159 undecided records. The DSU plot shows a more prominent classification decision boundary.

Specifically, in Figure 7a,b, the DSUs of the x and y axes are found at the two opposite ends of the same PC, as demonstrated in Figure 1. So, if one DSU captures the AV associations solely from class sepsis, the other DSU should capture the AV associations solely from class other. By observation, Figure 7b has a more prominent classification decision boundary than Figure 7a, suggesting that PC2’s clustering of the opposite groups of AV Associations is better aligned with the expected classification than PC1. The classification decision boundary can be curved as in Figure 7b, or approximated by a straight line as in Figure 7d. Both demonstrate the disentanglement capability of PDD.

The figures also revealed some insights about the abnormal records. First, some undecided records are close to the decision boundary, as shown in Figure 7a,b,d, since these records possess almost an equal number of patterns from both classes. This is more obvious in Figure 8b,d since the two classes are more disentangled and all the undecided records are highlighted. Second, most of the mislabelled records are consistently found in the sepsis cluster in Figure 7 and Figure 8. Specifically, 894 of the 996 (90%) mislabelled records were originally labelled as other but suggested as sepsis. This is more evident in Figure 8b,d since the two classes are more disentangled and all the mislabelled records are highlighted. Furthermore, all outlier records are situated close to the origin because they possess few patterns. Due to the unavoidable overlapping issue, this is more observable in Figure 7. Lastly, when the DSUs of both axes are associated with the same class but from different PCs, which is sepsis in Figure 7c and Figure 8c, the classification decision boundary is not as clear as in other figures when the DSU pairs are associated with different classes as in Figure 7a,b,d and Figure 8a,b,d.

## 5. Conclusions

In this study, we applied our Pattern Discovery and Disentanglement (PDD) methodology to healthcare data analysis, addressing challenges like mislabeling, class imbalance, and the need for explainable AI. The uniqueness and novelty of PDD lie in its ability to discover fundamental, explainable associations at the feature value level, unbiased by class labels, confounding factors, or imbalanced group sizes. PDD generates a comprehensive knowledge base that links patterns, entities, and sources/classes, providing deep insights and enhancing the interpretability of the results. After detecting the errors (i.e., outliers, undecided, and mislabelled), the original dataset can be cleaned. In this study, using the eICU-CRD data for sepsis risk assessment, PDD demonstrated significant improvements over traditional K-means clustering, achieving a 38% improvement on the full dataset and a 47% improvement on the reduced dataset. PDD’s error detection identified 996 mislabelled records, 2 outliers, and 159 undecided cases, resulting in the removal of 1157 abnormal records. This process improved the accuracy of multiple supervised ML models by an average of 4%, with improvements ranging from 1% to 9%. The analysis highlighted PDD’s ability to handle patterns challenging for traditional models, particularly lower-order patterns with limited feature correlation.

In conclusion, PDD offers a robust and practical solution for unsupervised clustering and error detection in large-size and low-quality healthcare data, enhancing both the stand-alone performance and the results of other ML models. Its application to this large, complex dataset demonstrates its practicality and effectiveness. Future work will extend PDD to other medical conditions and datasets, further validating its utility and impact in diverse healthcare applications.

## Figures and Tables

**Figure 1 bioengineering-11-00770-f001:**
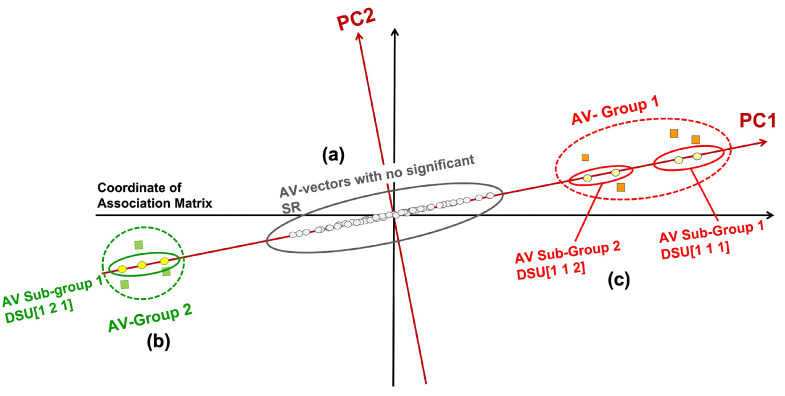
Principal Components (PC1, PC2) with AV Groups and Subgroups discovered by PDD. (**a**) On PC1, AVs with no significant statistical association with other AVs (SR less than the statistical significance threshold) are projected near the origin (marked as (**a**)). (**b**) AV Groups with statistically significant associations are projected further from the origin, indicating strong associations with other AVs (marked as (**b**)). (**c**) Subgroups, denoted by Disentangled Statistical Units (DSUs), with statistical associations among the AVs within but statistically uncorrelated to AVs of any other AV Subgroup (marked as (**c**)).

**Figure 2 bioengineering-11-00770-f002:**
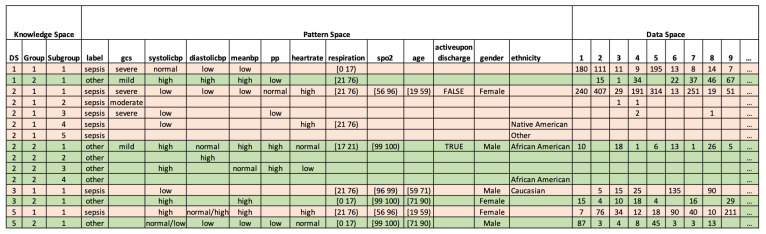
Knowledge Base outputted by PDD. Rows highlighted in pink and green represent patterns associated with the classes *sepsis* and *other*, respectively.

**Figure 3 bioengineering-11-00770-f003:**
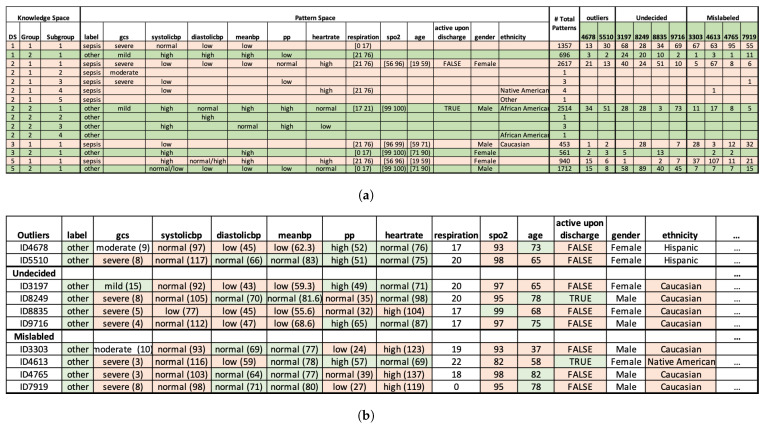
Error Detection results by PDD. (**a**) Knowledge Base of partial abnormal records. Rows highlighted in pink and green represent patterns associated with the classes *sepsis* and *other*, respectively. (**b**) Partial abnormal records of the original dataset. Blocks highlighted in pink and green represent feature values associated with the classes *sepsis* and *other*, respectively.

**Figure 4 bioengineering-11-00770-f004:**
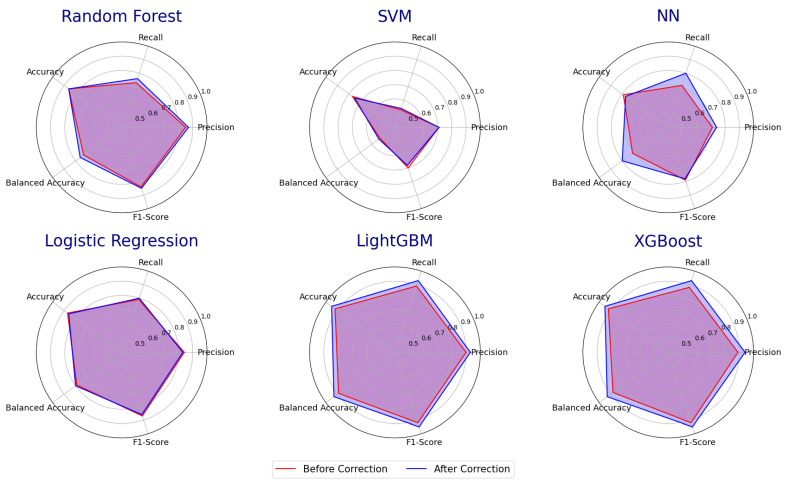
Comparison of the Accuracy of Classifiers Before and After Error Detection.

**Figure 5 bioengineering-11-00770-f005:**
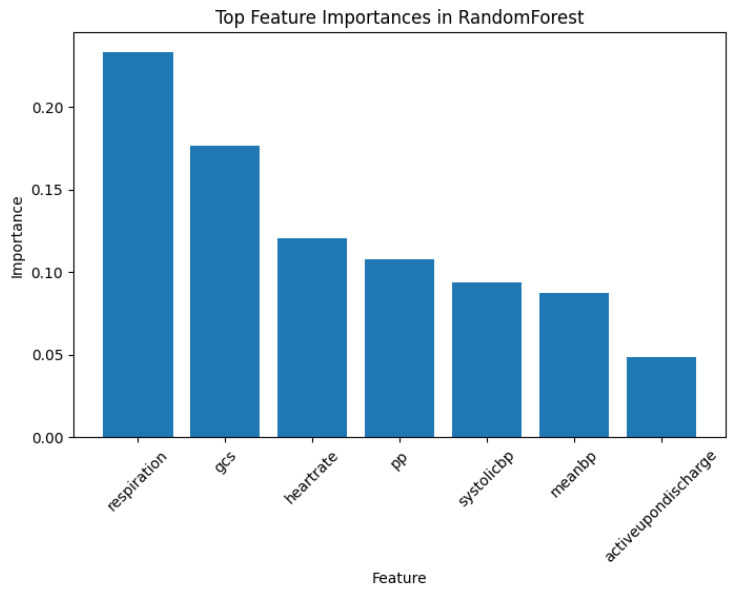
Top feature importances associated with class labels outputted by the Random Forest Model.

**Figure 6 bioengineering-11-00770-f006:**
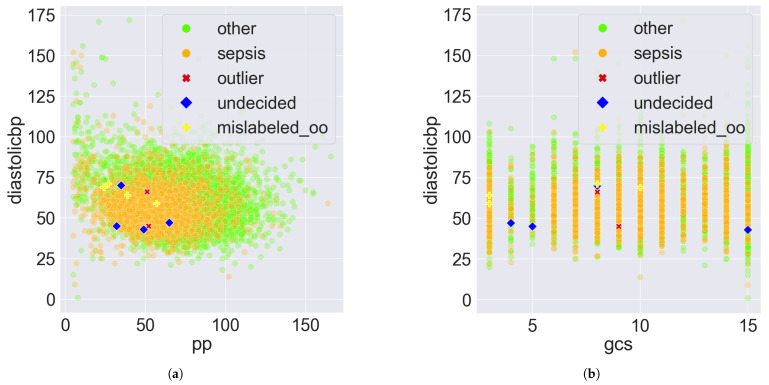
Sample scatter pair plots of the eICU-CRD dataset. All records are plotted. The 10 abnormal records presented in Section 3.3 are highlighted, which comprise 2 outliers, 4 undecided, and 4 mislabelled records originally labelled as other, denoted by mislabelled_oo. (**a**) diastolicbp vs. pp. (**b**) diastolicbp vs. gcs.

**Figure 7 bioengineering-11-00770-f007:**
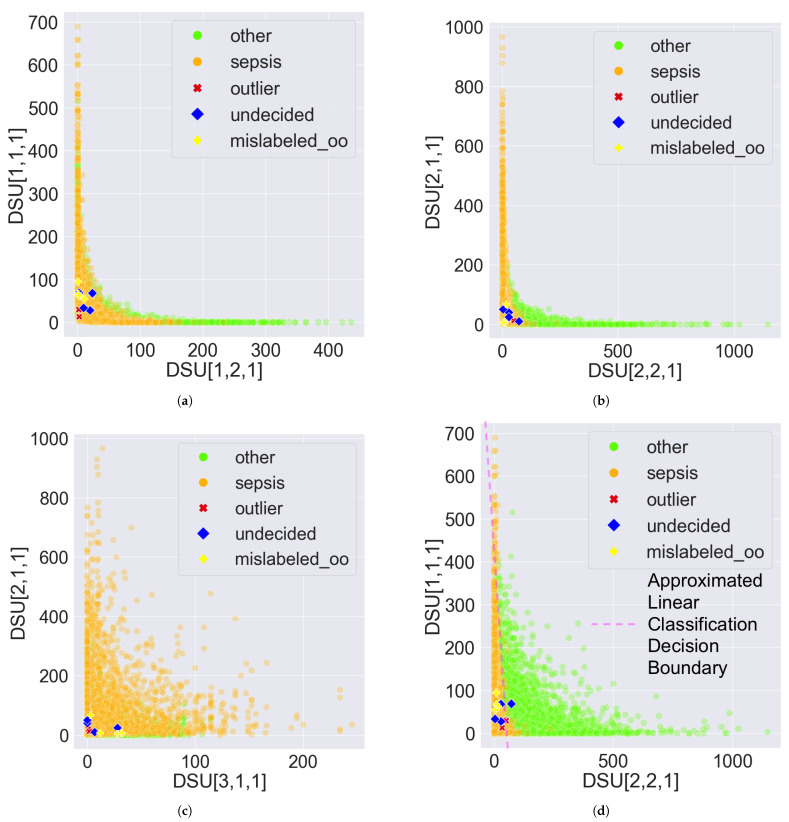
Sample scatter pair plots of the Data Space in the Pattern Discovery and Disentanglement (PDD)’s output. The Disentangled Space Units (DSU) in PDD are used as features, and the comprehensive pattern counts as the feature values. All records from the eICU-CRD dataset are plotted. The 10 abnormal records presented in Section 3.3 are highlighted, which comprise 2 outliers, 4 undecided, and 4 mislabelled records originally labelled as other, denoted by mislabelled_oo. (**a**) DSU[1,1,1] vs. DSU[1,2,1]. (**b**) DSU[2,1,1] vs. DSU[2,2,1]. (**c**) DSU[2,1,1] vs. DSU[3,1,1]. (**d**) DSU[1,1,1] vs. DSU[2,2,1].

**Figure 8 bioengineering-11-00770-f008:**
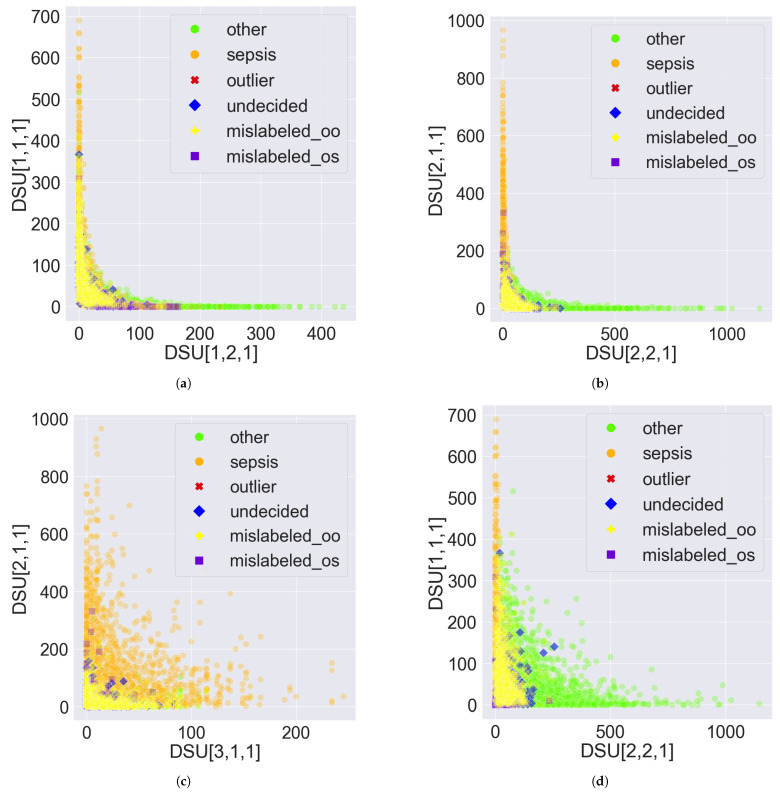
Sample scatter pair plots of the Data Space in the Pattern Discovery and Disentanglement (PDD)’s output. The Disentangled Space Units (DSU) in PDD are used as features, and the comprehensive pattern counts as the feature value. All records from the eICU-CRD dataset are plotted. All the 1157 abnormal records are highlighted, which comprise 2 outliers, 159 undecided and 996 mislabelled records. mislabelled_oo represents the 894 mislabelled records originally labelled as other, and mislabelled_os represents the 102 mislabelled records originally labelled as sepsis. (**a**) DSU[1,1,1] vs. DSU[1,2,1]. (**b**) DSU[2,1,1] vs. DSU[2,2,1]. (**c**) DSU[2,1,1] vs. DSU[3,1,1]. (**d**) DSU[1,1,1] vs. DSU[2,2,1].

**Table 1 bioengineering-11-00770-t001:** Clinician Feature Bins.

Features	Low Range	Normal Range	High Range
GCS	3–8 (Severe)	9–12 (Moderate)	13–15 (Mild)
Systolic BP ^1^	<90 mm Hg	90–120 mm Hg	>120 mm Hg
Diastolic BP ^1^	<60 mm Hg	60–80 mm Hg	>80 mm Hg
Mean BP ^1^	<70 mm Hg	70–93 mm Hg	>93 mm Hg
Pulse Pressure	<30 mm Hg	30–40 mm Hg	>40 mm Hg
Heart Rate	<60 bpm	60–100 bpm	>100 bpm

^1^ Blood Pressure.

**Table 2 bioengineering-11-00770-t002:** Comparison between PDD and K-means for unsupervised learning.

Unsupervised Learning	Precision	Recall	Accuracy	Balanced Accuracy	Weighted F1-Score
PDD (28-features data)	0.83	0.90	0.87	**0.89**	0.88
K-means (28-features data)	0.43	0.42	0.52	**0.42**	0.54
K-means (114-features data)	0.59	0.51	0.75	**0.51**	0.67

**Table 3 bioengineering-11-00770-t003:** Results of the Accuracy of Existing ML Models Before and After Error Detection Using PDD.

Classifiers (Before/After Error Detection)	Precision	Recall	Accuracy	Balanced Accuracy	Weighted F1-Score
Random Forest	0.85/0.87	0.73/0.76	0.86/0.86	**0.73/0.76**	0.84/0.85
SVM	0.71/0.71	0.53/0.54	0.77/0.75	**0.53/0.54**	0.7/0.68
NN	0.71/0.74	0.71/0.80	0.79/0.77	**0.71/0.80**	0.79/0.78
Logistic Regression	0.84/0.83	0.79/0.80	0.87/0.86	**0.79/0.80**	0.87/0.86
LightGBM	0.9/0.93	0.89/0.93	0.92/0.95	**0.89/0.93**	0.92/0.95
XGBoost	0.89/0.94	0.88/0.93	0.92/0.95	**0.88/0.93**	0.92/0.95

## Data Availability

Access to the eICU Collaborative Research Database can be requested at https://eicu-crd.mit.edu/ (accessed on 15 May 2023).

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
