# Peer review of "An Unsupervised Error Detection Methodology for Detecting Mislabels in Healthcare Analytics"

_bioengineering, 2024, doi:10.3390/bioengineering11080770_

Round 1

Reviewer 1 Report

Comments and Suggestions for Authors

This paper may seem like a valuable one, but due to its interdisciplinary nature and the author's overly concise introduction of many of its contents, it may seem difficult for readers including myself.

1. In the introduction, is there currently any relevant work foundation in this field?

2. In section 2.2.1, what is the basis for executing the work of "we removed redundant features"? Is there any standard? Otherwise, a subjective factor has been introduced here, which may affect the later results.

3. Equation 1, can you provide a specific example.

4. From lines 120 to 127, I don't quite understand. What is the author's work divided based on? Is it based on the quantity of data in each category, or on the actual meaning of data in each category?

5. Figure 1 is a crucial content that needs to be elaborated upon in detail.

6. Section 2.2.1 is crucial. Can you elaborate on it in more detail?

7. Bioengineering is an excellent journal in this field. Please learn more and cite relevant excellent papers published in it.

Reviewer 2 Report

Comments and Suggestions for Authors

THe title is suitable. The subject is of interest dor the journal. The abstract is well written. The introduction explain in more details the topic. The presented procedure is in some sense new and provides new ideas in the area. Due to the fact that the authors in more parts cited the recent developments of A.I and that they overcome some issue I suggest to include or to remark some aspect that in the prediction could be covered by genetic algoritms therefore it is suggeste to view to the following key point survey:

Transactions of the Institute of Measurement & ControlVolume 15, Issue 3, Pages 143 - 156August 1993 Document type Article Source type Journal ISSN 01423312 DOI 10.1177/014233129301500305  

Genetic algorithms and applications in system engineering: A survey

  • Caponetto R.;
  • Fortuna L.;
  • Graziani S.;
  • Xibilia M.G.

The results are suitable and well describe explain the improvements obtained.

In my opinion there are more innovative aspects respect the existing literature.

Author Response

Comments 1: The title is suitable. The subject is of interest dor the journal. The abstract is well written. The introduction explain in more details the topic. The presented procedure is in some sense new and provides new ideas in the area. Due to the fact that the authors in more parts cited the recent developments of A.I and that they overcome some issue I suggest to include or to remark some aspect that in the prediction could be covered by genetic algoritms therefore it is suggeste to view to the following key point survey:

Transactions of the Institute of Measurement & ControlVolume 15, Issue 3, Pages 143 - 156August 1993 Document type Article Source type Journal ISSN 01423312 DOI 10.1177/014233129301500305. Genetic algorithms and applications in system engineering: A survey. Caponetto R.; Fortuna L.;Graziani S.;Xibilia M.G.

The results are suitable and well describe explain the improvements obtained. In my opinion there are more innovative aspects respect the existing literature.

Response 1: Thank you for citing the Genetic Algorithm (GA) survey. We have carefully reviewed the recommended paper.

We agree that the Genetic Algorithm (GA) shares one similarity with PDD, which is codifying input. For instance, “A” and “B” from definition 1 of the GA survey on Page 144 resemble the concept of the “Address Table” introduced in our previous paper [2]. However, GA and PDD are different in their subsequent processes.

After codifying a given dataset, GA treats each record as a living individual. A generation starts by randomly selecting some records, where each record (parent) generates new records (offsprings) through operators such as cross-over and mutation, mimicking reproduction and gene inheritance. Parents are selected proportional to their fitness. New records are introduced into the original dataset by discarding an equal number of randomly chosen records. As pointed out by the GA survey, as long as the initial dataset is large enough, the original dataset will converge after many generations containing only the fittest genes [1]. Therefore, GA and PDD are fundamentally different. GA treats the dataset as a collection of possible solutions for a certain problem. GA modifies the dataset with a fitness bias, leading it to converge to a collection of optimal solutions. PDD does not modify the dataset. It extracts information from the dataset without bias and generates a knowledge base linking patterns, entities, and classes, which can be used to trace the origins of the discovered patterns.

Reference:
1. Caponetto, R., Fortuna, L., Graziani, S., & Xibilia, M. G. (1993). Genetic algorithms and applications in system engineering: a survey. Transactions of the Institute of Measurement and Control, 15(3), 143-156.

2. Wong, A. K., Zhou, P. Y., & Lee, A. E. S. (2023). Theory and rationale of interpretable all-in-one pattern discovery and disentanglement system. npj Digital Medicine, 6(1), 92.

To incorporate the above remarks, we have accordingly revised Section 1 – Introduction. A citation for the GA survey is added. The revised text is highlighted in red font.

Revised writing (from line 35 to line 44):

Next, healthcare providers often need clear explanations for AI-driven decisions, which is challenging with complex models like deep neural networks. Many of these models are black box models, such as genetic algorithms solely focusing on performance [10], lacking transparency and accountability, which can lead to severe consequences [11]. For high-stake applications such as healthcare, the judicial system, and human resource recruitment, machine learning models must be transparent to enable the interpretation of their predictions and explanation of decision boundaries [12] [11] [13]. As suggested by Rudin [11], creating an interpretable model is a better solution than trying to explain black box models. This allows for synergistic improvement as the interpretable model both integrates and enhances expert knowledge.

Round 2

Reviewer 1 Report

Comments and Suggestions for Authors

Authors have addressed all my concerns